# The Role of the Gut Microbiome in Liver Cirrhosis Treatment

**DOI:** 10.3390/ijms22010199

**Published:** 2020-12-28

**Authors:** Na Young Lee, Ki Tae Suk

**Affiliations:** Institute for Liver and Digestive Diseases, Hallym University College of Medicine, Chuncheon 24253, Korea; nylee@hallym.ac.kr

**Keywords:** liver cirrhosis, liver fibrosis, gut microbiome, gut-liver axis

## Abstract

Liver cirrhosis is one of the most prevalent chronic liver diseases worldwide. In addition to viral hepatitis, diseases such as steatohepatitis, autoimmune hepatitis, sclerosing cholangitis and Wilson’s disease can also lead to cirrhosis. Moreover, alcohol can cause cirrhosis on its own and exacerbate chronic liver disease of other causes. The treatment of cirrhosis can be divided into addressing the cause of cirrhosis and reversing liver fibrosis. To this date, there is still no clear consensus on the treatment of cirrhosis. Recently, there has been a lot of interest in potential treatments that modulate the gut microbiota and gut-liver axis for the treatment of cirrhosis. According to recent studies, modulation of the gut microbiome by probiotics ameliorates the progression of liver disease. The precise mechanism for relieving cirrhosis via gut microbial modulation has not been identified. This paper summarizes the role and effects of the gut microbiome in cirrhosis based on experimental and clinical studies on absorbable antibiotics, probiotics, prebiotics, and synbiotics. Moreover, it provides evidence of a relationship between the gut microbiome and liver fibrosis.

## 1. Introduction

Cirrhosis refers to scarring of liver tissue caused by long-term damage that prevents the liver from functioning properly. It is also called the end-stage of liver disease because it occurs after other stages of liver injury [1]. This can lead to serious, life-threatening complications such as bleeding, liver failure, or encephalopathy. There is currently no improved cure for liver cirrhosis. The only way is to manage symptoms and complications, in addition to slowing the progression of cirrhosis. If the liver is severely damaged, the only treatment option may be a liver transplant. The cost burden of cirrhosis treatment ranges from $14 million to $2 billion, depending on the cause of the disease [2].

Recently, several diseases have been found to be influenced by processes in the gut microbiome. Gut microbiome has also been implicated in interactions with certain drugs, including some psychiatric medications. Many studies have been performed to slow the progression of liver disease due to the modulation of the gut microbiome in nonalcoholic fatty liver disease [3,4]. These results showed that such changes in the gut microbial community can cause disorders in immune regulation which leads to disease.

Cirrhosis patients have altered gut-liver axis related to gut and systemic inflammation associated with changes in liver disease severity, damage to the gut barrier, and changes in the composition and function of gut microbiota [5]. Additionally, previous studies have reported that *Lachnospiraceae* and *Ruminococcaceae* are associated with the development of cirrhosis [6,7]. In addition to these changes, it is demonstrated that alteration in the function of bacteria which includes increased release of endotoxin and decreased conversion of primary bile acids to secondary bile acids may lead to cirrhosis [8]. Therefore, it can be surmised that the modulation of the gut microbiome plays an important role in the progression of cirrhosis.

Research on the relationship between dysbiosis and cirrhosis may not only predict the onset of cirrhosis but may also lead to discovery of novel treatments. Previous studies have demonstrated that microbiota targeted biomarkers can be a useful tool for the diagnosis of various diseases which includes liver cirrhosis [9]. Based on this, recent studies using antibiotics, probiotics, prebiotics, and synbiotics are being performed to suppress the progression of liver fibrosis by the modulation of the gut microbiome.

## 2. Liver Cirrhosis

Liver cirrhosis is defined as the late stage of liver fibrosis caused by several forms of liver disease and conditions, including hepatitis and chronic alcoholism [1]. It results from excessive production of extracellular matrix under chronic injury [10]. The mechanism of liver fibrosis is variable, depending on causes such as alcohol, hepatitis virus, or bile acids. The first step generally involves damage to liver cells by an injury that generates oxygen free radicals and inflammatory materials, following which Kupffer cells and inflammatory cells are activated and recruited. And then hepatic stellate cells are activated. This is the general mechanism of liver fibrosis [11,12,13]. Hepatic stellate cells, which occur in the space of Disse, play a major role in liver fibrosis [12].

Although liver fibrosis is a local reaction of the liver to chronic injury, serum levels of fibrogenic cytokines, extracellular matrix proteins, and degradation products are markedly increased in cases of advanced fibrosis (bridging fibrosis or cirrhosis) [14]. The matrix metalloproteinases (MMPs) and their inhibitors, tissue inhibitors of metalloproteinases, are proteins important in the matrix degradation. The finding that MMPs are expressed in hepatic injury indicates degradation of normal extracellular matrix may contribute to liver fibrosis. In a previous report, MMPs and microbiome are associated with fibrosis in respiratory disease [15]. MMPs play a central role in extracellular matrix remodeling in normal physiology. Each MMPs are associated with different stages of liver injury, including disease resolution, liver inflammation, fibrosis, cirrhosis and hepatocellular carcinoma.

The most common causes of cirrhosis are viral hepatitis (HV), nonalcoholic steatohepatitis (NASH), and alcoholic liver disease (ALD) [16]. Hepatocellular carcinoma (HCC) occurs in the background of a cirrhotic liver [17]. However, there is still no clear consensus for cirrhosis treatment Therefore, thus far, the goal of cirrhosis treatment is management of symptoms and complications.

In the past, the liver damage resulting from illness, excessive drinking of alcohol, or other cause was considered to be irreversible. Recently, however, many studies utilizing animal models provided evidence that cirrhosis may be reversible. In addition, some clinical studies have also shown the regression of cirrhosis on repeated biopsy samples [18]. Further research should be conducted in regards to therapeutic agents that may play a role in reversing cirrhosis.

## 3. Gut Microbiome

The gut microbiota in the human digestive tract consists of bacteria, protozoa, fungi, archaea, and viruses [19]. The gut microbiota is a complex ecosystem with a total mass of about 1–2 kg per person [20]. Most people have a population of bacteria in the gut that is about 10-factor number of cells in the body [21]. The gut microbiota is responsible for preventing and eliminating the invasion of pathogens, in addition to maintaining the balance of the immune system and preventing autoimmunity [22,23]. The gut microbiota is associated with essential health benefits, particularly in regards to immune homeostasis.

Birth and breastfeeding help form an infant gut microbiota that gradually matures in childhood in response to environmental exposure, after which the gut microbiota is relatively stable until changes in immune function leads to diminishing diversity [24]. Humans have an interdependent relationship with the gut microbiota. More than 90% of human gut microbiota consists of four major divisions: *Bacteroidetes*, *Firmicutes*, *Proteobacteria*, and *Actinobacteria* [25]. However, in those with compromised immune systems or with a progressive disease, the proportion and diversity of the intestinal microorganisms are different when compared to those of healthy people. In regards to this subject, animal models and patients of various diseases had their microbiota analyzed for composition and diversity [26].

In our previous study, we have demonstrated that mice on westernized diet were associated with decrease in *Bacteroidetes* and increased *Firmicutes* in their gut when compared to normal mice [3]. According to other studies, gut microbiota plays an important role in the host, which includes host immunity, food digestion, intestinal endocrine regulation, drug action and metabolism, and toxin elimination [27].

Metagenomic pyrosequencing of stool microbiota has given the way for the finding of novel genes from plenty microorganisms, and the analysis of whole genomes from community DNA sequence data [28]. Patients with cirrhosis revealed marked decrease in the functional genes involved in nutrient processing, including amino acids, lipids and nucleotides metabolism [28]. In a previous report, ammonia production and gamma-aminobutyric acid biosynthesis were enriched in patients with liver cirrhosis by their comparative metagenomic analysis with gene functional classification [9].

## 4. Gut-Liver Axis

The gut and liver communicate through tight bidirectional links through the biliary tract, portal vein, and systemic circulation [29]. The close relationship between the gut and liver underlies the modulatory effect of gut microbiota on liver health [30]. Moreover, dysbiosis, which refers to quantitative and qualitative changes in gut microbiota and its overgrowth, may lead to an increase in intestinal permeability. As a result, endotoxins are transferred to the portal vein, leading to the activation of signaling pathways of various inflammatory cytokines in the liver [20]. Microbial products serving as pathogen-associated molecular patterns bind to toll like receptor (TLR) and activate the Kupffer cells, stimulating innate immune responses, including inflammatory cytokines. Various toxin binds to TLR4 and activates TLR9 [31]. TLR2 is bound by lipoprotein or peptidoglycan of Gram (+) microbiota [32]. Plasma endotoxin levels are elevated with the progression of liver cirrhosis [33].

Bile acid is the important factor in the axis between the liver and the gut. The main primary bile acids are synthesized by the liver and are combined with taurine or glycine to be secreted into bile. Thereafter, the synthesized bile is stored in the gallbladder and delivered to the small intestine. Gut microbiome generate secondary bile acids, including deoxycholic acid and lithocholic acid, by deconjugation and dihydroxylation and are reabsorbed into the enterohepatic circulation at the ileum. bile acids are important not only for the absorption of vitamins and dietary fats but also as ligands for the nuclear receptor farnesoid X receptor and the Takeda-G-protein coupled receptor [34,35]. Therefore, the close interaction between the gut and the liver can be a major factor in the pathogenesis of liver damage and liver disease progression.

## 5. Dysbiosis and Bacterial Translocation

Dysbiosis is a term for a microbial imbalance or maladjustment inside the body. This state of imbalance can be caused by changes in the number of microbes in a particular population that can have a profound influence on energy and immune homeostasis, which result in significant metabolic and immunologic effects on the host [36]. This persistent imbalance of gut microbial community is accompanied by a wide range of systemic symptoms of gastrointestinal diseases such as inflammatory bowel disease and irritable bowel syndrome, obesity, type 2 diabetes, and atopy [37]. As such, the imbalance of the microbiota can serve as an indicator for pathological state.

Dysbiosis is associated with gut barrier dysfunction and immunity since the microbiota and its products modulate barrier function by affecting the epithelial inflammatory response and mucosal repair function [38]. Previous studies have shown that cirrhosis is associated with altered immune responses that potentially allows dysbiosis or altered microbiota in the stool, intestinal mucosa, ascites, liver, serum, and saliva [39]. For this reason, many studies are being performed to investigate the possibility of alleviating cirrhosis by modulating the gut microbiome. Dysbiosis of the gut is related with various human diseases such as obesity [40], metabolic disease [41], diabetes mellitus [42], vascular disease [43], chronic liver disease [44,45] and neuroinflammatory disease [46]. Neuropsychologically, bidirectional interaction (gut-brain axis) between brain and gut microbiome via neurological or immunological mechanisms is closely related with dysbiosis.

Bajaj et al. suggested that the recto-sigmoid mucosa-microbiota in cirrhosis revealed a lower abundance of autochthonous bacteria (*Subdoligranulum*, *Dorea*, and *Incertae Sedis* XIV other) and a higher abundance of potentially pathogenic bacteria (*Enterococcus*, *Clostridium*, *Burkholderia*, and *Proteus*) [47]. Other study demonstrated that *Veillonella*, *Megasphaera*, *Dialister*, *Atopobium* and *Prevotella* were increased in cirrhotic patients [6].

Theoretically, considering the recovery function of the intestinal microflora to a healthy state, the modulation of dysbiosis might be considered as a potential therapeutic option for treating liver cirrhosis. This hypothesis has been recently demonstrated by some strong evidences [48,49,50,51,52] (Figure 1).

In fibrosis and cirrhosis, intestinal dysbiosis, gut barrier dysfunction, and systemic immunologic dysfunction cause bacterial translocation [53]. Bacterial translocation, is defined as the migration of viable intestinal microorganisms or their products to the mesenteric lymph nodes or other sites. The liver is a central immunological organ that is composed with innate immune cells and constantly exposed to circulating endotoxins derived from intestinal microbiome [54]. Intestinal immune dysregulation due to intestinal immune system abnormalities is main event in patients with cirrhosis. With cirrhosis progression, intestinal immune dysregulation and gut dysbiosis worsened.

## 6. Treatment for Cirrhosis

The liver is the organ that metabolizes and detoxifies various compounds. Therefore, toxicity from the most common and serious drug should be considered. Therefore, recent studies are trying to find a treatment for liver disease using pharmabiotics. The main approach in the treatment of liver cirrhosis is anti-fibrosis therapy that targets the liver fibrosis-generating mechanism irrespective of the cause of cirrhosis. In order to develop effective treatments for liver fibrosis, the molecular pathogenesis and treatment of liver fibrosis have been exhaustively investigated over the last two decades. As a result, some candidates with anti-fibrotic effects in animal experiments have been found; however, most of these have not been verified for use in human beings. Recently, since intestinal microbes have been identified as the cause of liver disease, treatment using intestinal microbes has emerged. Summarizing the reported data, modulation-related therapeutic effects were mostly associated with inflammation, and the use of pharmabiotics has shown improvement in inflammatory and immune mechanisms.

### 6.1. Cirrhosis and Antibiotics

People with cirrhosis, especially those with decompensated cirrhosis, have an increased risk of bacterial infection, which can further promote other hepatic decompensation, including liver failure [55]. In theory, antibiotics may eliminate deleterious bacteria and their efficacy in treating liver disease has been proven in research [56]. For this reason, many studies are being performed to investigate antibiotics treatment in the context of cirrhosis (Table 1). Traditional antibiotics may not be effective in controlling microbiota due to side effects and the emergence of antibiotic resistance [57]. Nevertheless, treatment with rifaximin has shown promising results in relieving cirrhosis while modulating gut microbiota. Furthermore, ingestion of probiotics can ameliorate cirrhosis of the liver, in addition to many immune effects involving various cytokines such as IL-6, TNF-a, and IL-1B. Rifaximin is a gastrointestinal selective antibiotic with a wide range of antimicrobial activity, minimal drug interactions and negligible effect on the overall gut microbiome [58] (Table 1). In a study in EtOH-induced liver injury in obese mice, treatment with rifaximin increased proportion of the *Bacteroidales* and decreased alanine aminotransferase (ALT) and triglycerides (TG) levels via modulation of small intestine [59].

Most treatments for hepatic encephalopathy patients rely on manipulation of the intestinal environment, so antibiotics acting on the gut represent the main treatment strategy [60]. In previous research, rifaximin treatment effects were shown to decrease the risk of recurrent encephalopathy [61]. Patients with cirrhosis who developed candidemia also were shown with a lower rate of candidemia when treated with rifaximin [62]. Cirrhotic patients with refractory ascites when given rifaximin were associated with mitigated ascites and increased survival [63]. As such, rifaximin may be effective but how it affects these therapeutic outcomes remains unknown. It is not yet clear to what extent antibiotics can control the composition and diversity of gut microbiota in a variety of clinical settings. Another study showed that rifaximin-α treatment had no effects on macrophage activation and disruption of fibrosis [64]. Another previous study has shown that in cirrhotic patients, treatment by rifaximin reduced *Veillonellaceae* and secondary/primary BA ratios [65]. It suggests that cirrhosis is associated with a reduced conversion of primary to secondary Bas, which is associated with the abundance of major gut microbiota such as *Enterobacteriaceae*, *Lachonospiraceae*, *Ruminococcaceae* and *Blautia*. In a randomized controlled trial of patients with advanced cirrhosis, treatment with norfloxacin did not reduce mortality, but significantly reduced the incidence of Gram-negative bacterial infections without increasing infections due to multiple resistant bacteria [66].

Antibiotics have a significant direct or indirect effect on the intestinal microbiota and some changes disappear immediately after stopping antibiotic treatment, but others remain indefinitely [67]. Antibiotics for cirrhosis prevent bacterial infection and other cirrhosis complications such as recurrent varicose bleeding and death. However, their widespread use has led to development of antibiotic resistance, which makes standard empirical antibiotics for suspected infections ineffective and perhaps reduces the effectiveness of antibiotic prophylaxis. To prevent the occurrence of antibiotic resistance, an empirical antibiotic strategy, step-down rules, and antibiotic pharmacokinetics, and pharmacodynamic administration strategies should be formulated.

Moreover, antibiotics affect the bacteria that cause infections as well as the resident microbiota [68]. Antibiotics are used with the intention of removing pathogenic bacteria and inhibiting proliferation, but due to their broad-spectrum activity, they can indiscriminately kill or inhibit a subset of symbiotic microbes [69]. It may result in the decrease of taxonomic richness, diversity, and evenness of the community. Although this side effect has long been appreciated, advances in sequencing technologies enable a detailed study of how antibiotics alter the gut microbiome. Direct effects on the immune system, reproducibility in terms of duration and frequency of antibiotic exposure, antibiotic resistance, and individualized response to the same treatment all influence the outcome of antibiotic studies. It is necessary to develop strategies to mitigate the effects of antibiotics on the immune system. 

### 6.2. Cirrhosis and Probiotics

Probiotics are defined as live microorganisms such as bacteria or yeasts of human origin that provide health benefits when consumed [71]. Many studies have been conducted on patients with non-alcoholic fatty liver disease and irritable bowel syndrome using different types of probiotics in different settings [72,73] (Table 2). Current studies have shown that probiotics regulates the gut microbiota by promoting the growth of beneficial bacteria and reducing harmful bacteria in the gut [74,75].

Bile duct ligation (BDL) is a surgical method that is used to induce liver fibrosis. It leads to acute progress to cirrhosis with portal fibrosis [76]. In the BDL model, BA de novo synthesis was decreased after the administration of *Lactobacillus rhamnosus GG*. In addition, the administration resulted in the reduction of aspartate aminotransferase (AST), ALT, alkaline phosphatase, and total bilirubin (TBIL) serum levels and α-SMA, Col1, Col3, and transforming growth factor (TGF)-β mRNA levels [77]. Carbon tetrachloride (CCl_4_) is typically used to create models of liver fibrosis and cirrhosis [78]. In a study using CCl_4_ injection to induce liver cirrhosis in mice, mixture of *S. cerevisiae* and *L. acidophilus* protected mice from inflammation, hepatic oxidative stress by reducing MAPK signaling and increasing SIRT1 signaling [79]. In another study, *L. fermentum* and *L. plantarum* administration was associated with reduced AST, ALT, MDA, SOD, GSH, and interleukin (IL)-1β levels; in contrast Bcl-2 was increased [80]. In rats CCl_4_ injection and treatment with *L. salivarius LI01*, and *Pediococcus pentosaceus LI05* were associated with reduction of the collagen type 1a (Col1a), tissue inhibitor of metallopeptidase 1 (Timp1), and TGF-β when compared with the control group. Moreover, these strains increased the expression of tight junction protein Zo-1 [81]. In the EtOH-induced model, combination with *L. fermentum* resulted in reduced AST, ALT, iNOS, and Hsp60 [82]. This study suggests probiotics are associated with therapeutic potential in alcoholic liver disease. Mixture of *L. paracasei*, *L. casei*, and *Weissella confusa* treatment significantly lowered serum enzyme levels, less inflammation, and less fibrosis on TAA-induced liver fibrosis in rats [83]. In this experiment, rats were fed 10^9^ CFU/mL microbial cells daily by oral gavage.

In a human study, randomized patients were given VSL#3 probiotics for 6 months. Patients who received probiotics were associated with decreased hepatic encephalopathy incidence, and child–pugh score (CTP) and model for end-stage liver disease (MELD) scores were also reduced [84]. In another study, *Bifidobacterium breve*, *L. acidophilus*, *L. plantarum*, *L. paracasei*, *L. bulgarius* and *Streptococcus thermophilus* treatment of patients with hepatic encephalopathy improved CTP score and psychometric hepatic encephalopathy scores [85]. In clinical trials, the effects of treatment with the probiotic *L. rhamnosus GG* in patients with cirrhosis were evaluated [86]. This study showed that ingestion of *L. rhamnosus GG* decreased *Enterobacteriaceae*, endotoxemia and tumor necrosis factor (TNF)-α. The fecal microbiome composition of *Lachnospiraceae* was increased and harmful bacteria was reduced after ingestion of *L. rahmnosus GG.* The study concluded that *L. rhamnosus GG* modulates the gut microbiome, metabolome and endotoxemia in cirrhosis patients. In another study, treatment by *C. butyricum* combined with *B. infantis* in minimal hepatic encephalopathy in hepatitis B virus-induced cirrhosis patients were associated with decreased *Enterococcus*, *Enterobacteriaceae*, and ammonia levels [87]. Moreover, their cognitive ability was improved.

Subsequently, various studies with patients and animal models of liver fibrosis are aiming to investigate the improvement of liver fibrosis and cirrhosis following the ingestion of probiotics. However, well-designed long-term clinical trials with probiotics are required to assess the probiotics’ effects on the progression of liver disease and regression of liver fibrosis. Further research to elucidate the mechanism underlying the role of probiotics in modulating the gut microbiome study is required. 

### 6.3. Cirrhosis and Prebiotics

Prebiotics were first identified and defined as an indigestible food ingredient that has a beneficial effect on improving host health by selectively stimulating the growth or activity of bacteria by Roberfroid and Gibson in 1995 [88]. Namely, prebiotics are food ingredients that induce the growth or activity of beneficial microorganisms in the gut. The food ingredients can feed the gut microflora, and the products of their breakdown such as short chain fatty acids are released into the blood circulation, which affect not only the gastrointestinal tract, but also other distant organs [89]. Sources of prebiotics include breast milk, soybeans, inulin, raw oats, unrefined wheat, unrefined barley, yacon, undigestible carbohydrates, and undigestible oligosaccharides [90].

Garlic consumption is known to be beneficial in various liver diseases [91]. In one study, ingestion of garlic polysaccharides reduced the ratio of AST, ALT, TGF-B1, and TNF-a in the acute liver failure model [87] (Table 3). In addition, garlic polysaccharides affected the gut microbiota, resulting in an increase in *Lachnospiraceae* and *Lactobacillus* and a decrease in *Facklamia* and *Firmicutes*. With the CCl4-injected mice, which is a widely known liver cirrhosis model, studies on intake of various prebiotics have been performed. Ingestion of polysaccharides from *Grifola frondosa* decreased AST, ALT, TBIL, MDA, TNF-α, IL-1β, and IL-6, and increased SOD and GSH-Px [92]. In this study, it was reported that polysaccharides from *Grifola frondosa* inhibited oxidative stress and inflammatory reactions to regulate the TGF-β1 / Smad signaling pathway and slow the progression of liver fibrosis. In addition, the ingestion of *Dendrobium officinale* polysaccharide was shown to alleviate fibrosis tissue and reduce intestinal mucosa damage [93]. Because the expression of Bax and caspase-3 proteins was downregulated, the expression of occludin, claudin-1, zonula occludenes-1 and Bcl-2 proteins was upregulated. In another study, olive oil combined with *Lycium barbarum* polysaccharide improved hepatocellular death, inflammation, and fibrosis markers in liver cirrhosis induced rat model [94]. It was shown that TGF-β1, TNF-α, and Timp-1 were decreased, and IL-10, IL-10/TNF-α were increased. Inulin is an indigestible storage polysaccharide that is found in many vegetables [95]. In the alcohol animal model, inulin was shown to increase intestinal content of propionic acid, butyric acid and valeric acid [96]. Short chain fatty acids with a small number of carbon atoms, such as propionic acid and butyric acid, are partially absorbed and are reported to inhibit cholesterol synthesis in the liver and promote the decomposition of low density lipoprotein cholesterol [97].

Non-absorbable disaccharides are recommended as the main treatment for hepatic encephalopathy since their beneficial effects involve the reduction of the intestinal production and absorption of ammonia [98]. Lactitol, one of the non-absorbable disaccharide, is a crystalline powder sweetener similar in sweetness to sugar [99]. In a previous study, a randomized clinical trial was performed in which people with cirrhosis and hepatic encephalopathy were given lactitol [100]. Although there were no statistical differences between randomized clinical trials when evaluating hepatic encephalopathy, lactitol intake was associated with beneficial effects on the quality of life. It also had beneficial effects on mortality versus placebo.

A number of studies continue the investigation of new polysaccharides for the development of effective treatments for liver damage and liver disease [101]. Further research on the mechanism by which probiotics can play a role as a therapeutic agent for liver cirrhosis by modulating the gut microbiome is further needed.

### 6.4. Cirrhosis and Synbiotics

Synergistic combinations of probiotics and prebiotics are defined as synbiotics [90]. Synbiotics were developed to overcome possible survival difficulties for probiotics. As well as promoting the growth of probiotics and bacteria, synbiotics contribute to a more efficient homeostasis in the gut and maintenance of a healthy body [102].

In a previous study, hepatic encephalopathy patients were treated with a mixture of four probiotics (*L. paracasei* + *L. plantarum* + *L. mesenteriodes* + *P. pentosaceus*) in combination with four fibers (oat bran, pectin, resistant starch, and inulin) [103] (Table 4). There were no significant differences between randomized groups at baseline. In cirrhosis patients with minimal hepatic encephalopathy and not overt hepatic encephalopathy showed that treatment of synbiotics which consisted of mixture of four probiotics (*P. pentoseceus* + *L. mesenteroides* + *L. paracasei* + *L. plantarum*) with three fibers (beta glucan + pectin + resistant starch) was associated with decreased TBIL levels in serum and increased albumin levels [104].

## 7. Cirrhosis and Gut Microbiome

The liver is the organ that is in closest contact with the gut tract and is exposed to a substantial number of bacterial components and metabolites. Previous studies have proposed that microbiota-based biomarkers may be a tool to diagnose cirrhosis [9]. For example, cirrhosis patients have increased bacteremia, increased levels of serum lipopolysaccharides, and increased intestinal permeability [105].

Alcoholic liver disease is caused by various factors which includes genetics, immune system, dietary components, and the gut microbiota. In a previous study, a subgroup of alcoholic liver disease patients exhibited dysbiosis with lower median abundances of *Bacteroidetes* and higher levels of *Proteobacteria* [106]. Certain microorganisms can induce alcoholic liver disease, while others can exert beneficial effects and have protective effects. In addition, in cirrhosis patients, *Bacteroidetes* were shown to be decreased significantly, while *Proteobacteria* and *Fusobacteria* were increased significantly when compared to healthy people [6]. These findings suggest the important role of gut microbiome in patients with cirrhosis.

The major role of the gut microbiota in liver disease is also supported by various studies showing that several complications of serious liver disease, such as hepatic encephalopathy are efficiently treated by the modulation of gut microbiome via use of probiotics, prebiotics and antibiotics [107]. The pathogenesis of cirrhosis and the precise function of gut microbiome are not yet clear, but these findings that improvement in liver cirrhosis-induced animals and patients highlight the importance of modulation of the gut microbiome, suggesting novel approaches for therapeutic strategies for liver fibrosis. The modulation of gut microbiota with a healthy diet that helps gut microbial activity such as fiber-based, multi-biotics based supplements, and transplantation of a fecal microbiome from healthy subjects to promote the growth of “good” microbiota may ameliorate dysbiosis in patients and improve their prognosis [108].

Most of the studies used 16S rRNA sequencing, and detailed analysis up to the genus level was possible. Future studies should focus on metagenomics where we can reach up to species-level of microbes by using total DNA sequencing (shotgun metagenomics). And then, we can identify and characterize biological mechanisms that drive the human response to an intervention of pharmabiotics.

## 8. Conclusions

The gut-liver axis plays an important role in the pathogenesis of liver diseases, including liver fibrosis and cirrhosis. Chronic liver disease, especially fibrosis and cirrhosis, are serious disease with many side effects. The fact that liver cirrhosis is related to the microbiome and the possibility that it can be treated by controlling the microbiome is expected to affect the development and health improvement in the medical field in the future. Therefore, it is necessary to evaluate the manipulation of the intestinal microbiota in the context of liver cirrhosis

Consequently, a comprehensive understanding of the pathology of liver cirrhosis is important for improving clinical outcomes, as integrated signaling pathways appear to play an important role in pathogenesis of liver cirrhosis. Further studies are needed to study the interaction between gut microbes and the host immune system in order to elucidate the pathogenesis of liver fibrosis and open new opportunities in immunity or gut microbiome-based treatments. Future trials of probiotics, prebiotics, and synbiotics are recommended to include metagenomic and/or metabolomic analysis for evaluation of their effects and their possible backgrounds.

## Figures and Tables

**Figure 1 ijms-22-00199-f001:**
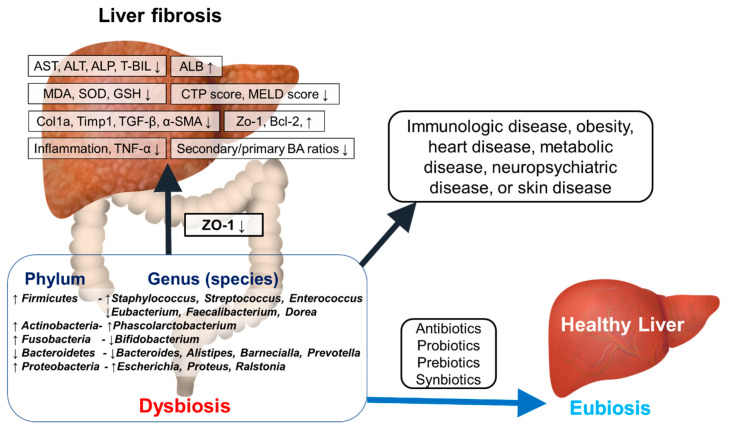
Dysbiosis and diseases. AST, aspartate aminotransferase; ALT, alanine aminotransferase; T-BIL, total bilirubin; ALB, albumin; MDA, malondialdehyde; SOD, superoxide dismutase; GSH, Glutathione; CTP, Child-Turcotte-Pugh; MELD, model for end-stage liver disease; Col, Collagen; Timp, tissue inhibitor of metallopeptidase; TGF, transforming growth factor; α-SMA, alpha-smooth muscle actin; TNF- α, tumor necrosis factor alpha; Zo-1, zonula occludenes-1; BA, bile acids; BCL-2, b-cell lymphoma 2.

**Table 1 ijms-22-00199-t001:** Animal and human studies using antibiotics.

Conditions		Treatment	Main Results	Ref
Animal	EtOH-induced liver injury in obese mice	Rifaximin	(↓): ALT, TG (↑): Proportion of the *Bacteroidales*	[59]
Human	HE	Rifaximin	(↓): Recurrent encephalopathy	[61]
	Cirrhosis developing candidemia	Rifaximin	(↓): Rate of candidemia	[62]
	Cirrhotic patients with refractory ascites	Rifaximin	(↑): Ascites and survival of cirrhotic patients	[63]
	Cirrhosis	Rifaximin-α	No effects on macrophage activation and disruption of fibrosis	[64]
	Cirrhosis	Rifaximin	(↓): *Veillonellaceae*, secondary/primary BA ratios	[65]
	Advanced cirrhosis	Norfloxacin	(↓): Incidence of Gram-negative bacterial infection (↑): Survival of patients with low ascites protein concentration	[66]
	Cirrhosis	Poorly/non-absorbable antibiotics	(↓): Hepatic venous pressure gradient	[70]
	HE (review)	Antibiotics Rifaximin	Improve cognition, inflammation, quality -of-life and driving simulator performance	[60]

↑ indicates an increase in condition, ↓ indicates a decrease in condition, ALT, alanine aminotransferase; BA, bile acids; HE, hepatic encephalopathy; TG, triglycerides

**Table 2 ijms-22-00199-t002:** Animal and human studies using probiotics.

Conditions		Treatment	Main Results	Ref
Animal	BDL	*L. rhamnosus GG*	(↓): BA de novo synthesis, ALT, AST, ALP, TBIL, a-SMA, Col1, Col3, TGF-β, Timp1, Mmp2, F4/80, TNF-α, IL-6, IL-1B (↑): FGF-15, BA excretion	[77]
	CCl_4_	Mixture of *Saccharomyces cerevisiae* + *L. acidophilus*	(↓): hepatic oxidative stress, ER stress, inflammation, MAPK signaling, AST, ALT, Col1, α-SMA (↑): SIRT1 signaling	[79]
		*L. fermentum*	(↓): Inflammation, AST, ALT, MDA, SOD, GSH, IL-1β, Bax, TNF-α, Caspase 3↓, NF-κB, p65 (↑): Bcl-2	[80]
		*L. plantarum*	(↓): ALT, AST, MDA, SOD, GSH, IL-1β, TNF-α, Bax, NF-κB p65, Caspase (↑): Bcl-2	
		*L. salivarius LI01*	(↓): AST, ALT, GGT, TLR2,4,5,9, intestinal barrier integrity, Col1a, Timp1, TGF-B (↑): Zo-1	[81]
		*P. pentosaceus LI05*	(↓): AST, ALT, GGT, TLR2,4,5,9, Col1a, Timp1, TGF-β(↑): Zo-1	
	EtOH	*L. fermentum*	(↓): steatosis score, iNOS, Hsp60, AST, ALT	[82]
	TAA	Mixture of *L. paracasei* + *L. casei* + *W. confusa*	(↓): serum enzyme levels, inflammation, fibrosis	[83]
Human	Cirrhosis with HE	VSL #3: *L.* (*acidophilus + delbrueckii subspbulgaricus + casei + plantarum*) *+ Bifidobaceria* (*breve + longum + infantis*) *+ S. salivarius subspthermophilus*	(↓): CTP score, MELD score, IL-1β, IL-6, TNF-α, Indole, Renin, Aldosterone, Brain-type natriuretic peptide	[84]
	Cirrhosis without overt HE	*B. breve*, *L. acidophilus*, *L. plantarum*, *L. paracasei*, *L. bulgaricus*, *S. thermophilus*	(↓): CTP score, psychometric hepatic encephalopathy scores	[85]
	Cirrhotic with MHE	*L. rhamnosus GG*	(↓): *Enterobacteriaceae*, endotoxemia, TNF-a (↑): *Clostridiales*, *Lachnospiraceae* relative abundance	[86]
	Minimal MHE in HBV-induced Cirrhosis	*Clostridium butyricum + B. infantis*	(↓): *Enterococcus*, *Enterobacteriaceae*, ammonia level (↑): Cognitive ability	[87]

↑ indicates an increase in condition, ↓ indicates a decrease in condition, ALT, alanine aminotransferase; ALP, alkaline phosphatase; AST, aspartate aminotransferase; α-SMA, alpha-smooth muscle actin; BA, bile acids; Bax, Bcl-2-associated X protein; BCL-2, b-cell lymphoma 2; BDL, bile duct ligation; CCl4, carbon tetrachloride; Col; Collagen, type; CTP, Child-Turcotte-Pugh; EtOH, ethylalcohol; GGT, gamma glutamyl peptidase; GSH, Glutathione; HBV, hepatitis B virus; HE, hepatic encephalopathy; HSP, heat shock proteins; IL, interleukin; iNOS, inducible nitric oxide synthase; MAPK, mitogen-activated protein kinase; MDA, malondialdehyde; MELD, model for end-stage liver disease; MHE, minimal hepatic encephalopathy; Mmp, matrix metallopeptidases; NF-κB, nuclear factor kappa-light-chain-enhancer of activated B cells; SIRT, selective internal radiation therapy; SOD, superoxide dismutase; TAA, thioacetamide; TG, triglycerides; TGF, transforming growth factor; TBIL, total bilirubin; Timp, tissue inhibitor of metallopeptidase; TLR, toll-like receptor; TNF- α, tumor necrosis factor alpha; Zo-1, zonula occludenes-1

**Table 3 ijms-22-00199-t003:** Animal and human studies using prebiotics.

Conditions		Treatment	Main Results	Ref
Animal	ALF	Garlic polysaccharide	(↓): AST, ALT, MDA, TC, TG, TGF-β1, TNF-α, *Lachnospiraceae*, *Lactobacillus* (↑): SOD, GSH-Px, GSH, *Firmicutes*, *Facklamia*	[87]
	CCl_4_	Polysaccharides from *Grifola frondosa*	(↓): AST, ALT, TBIL, MDA, TNF-α, IL-1β, IL-6 (↑): SOD, GSH-Px	[92]
		*Dendrobium officinale* polysaccharide	(↓): Bax, caspase 3, TNF-α, α-SMA (↑): occludin, claudin-1, ZO-1, Bcl-2 TEER, IL-10	[93]
		Olive oil combined with *Lycium barbarum* polysaccharides	(↓): TGF- β1, TNF-α, Timp-1 (↑): IL-10, IL-10/TNF-α	[94]
	ALD	Inulin	(↓): iNOS, inflammation, TNF-α (↑): propionate, butyrate, valeric, IL-10	[96]
Human	Cirrhosis with HE	Lactitol	(↓): Mortality	[100]

↑ indicates an increase in condition, ↓ indicates a decrease in condition, ALD, alcoholic liver disease; ALF, alcoholic liver fibrosis; ALT, alanine aminotransferase; AST, aspartate aminotransferase; α-SMA, alpha-smooth muscle actin; Bax, Bcl-2-associated X protein; CCl_4_, carbon tetrachloride; GSH, glutathione; HE, hepatic encephalopathy; IL, interleukin; MDA, malondialdehyde; SOD, superoxide dismutase; TC, total cholesterol; TG, triglycerides; TGF, transforming growth factor; TBIL, total bilirubin; TNF-α, tumor necrosis factor-alpha

**Table 4 ijms-22-00199-t004:** Human studies using synbiotics.

Conditions		Treatment	Main Results	Ref
Human	HE	Mixture of 4 probiotics (*L. paracasei* + *L. plantarum + Leuconostoc mesenteriodes* + *P. pentosaceus*) with 4 fibers (oat bran, pectin, resistant starch, and inulin)	No change in cognitive function	[103]
	Cirrhosis with MHE	Mixture of 4 probiotics (*P. pentoseceus* + *Leuconostoc mesenteroides* + *L. paraacasei + L. plantarum*) with 3 fibers (beta glucan + pectin + resistant starch)	(↓): TBIL (↑): ALB	[104]

↑ indicates an increase in condition, ↓ indicates a decrease in condition, ALB, albumin; HE, hepatic encephalopathy; MHE, minimal hepatic encephalopathy; TBIL, total bilirubin

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
