# Peer review of "The Role of the Gut Microbiome in Liver Cirrhosis Treatment"

_ijms, 2020, doi:10.3390/ijms22010199_

Round 1

Reviewer 1 Report

This review should be renamed " The role of the gut microbiome in liver cirrhosis treatment ".

This title is more appropriate as the pathophysiology section is very small and, perhaps, devoid of this manuscript. 

They are appreciable the evidences on gut microbiota modulation in liver cirrhosis. Tables help a lot the reader to have a brief view of the cited studies. 

A mor striking Conclusions section is needed: it cannot be a summary of the reported evidences. We want the authors to resemble the evidences, discuss on them and write about future challenges or roads to run! 

A fair English revision is neded. 

Author Response

ijms-1041366

“The modulation of the gut microbiome in liver fibrosis”  

Point-to-point responses to comments by the Reviewer 1.

Major Criticism

  • Comment 1: This review should be renamed "The role of the gut microbiome in liver cirrhosis treatment". This title is more appropriate as the pathophysiology section is very small and, perhaps, devoid of this manuscript.

  • Response 1: Thanks for the advice of it. We also considered what was suggested in the title. We will modify it as you suggested.

  • Comment 2: They are appreciable the evidences on gut microbiota modulation in liver cirrhosis. Tables help a lot the reader to have a brief view of the cited studies. A more striking Conclusions section is needed: it cannot be a summary of the reported evidences. We want the authors to resemble the evidences, discuss on them and write about future challenges or roads to run! 

  • Response 2: We appreciate the reviewer’s thoughtful comment. We changed conclusion section with future challenges or roads to run.

“The gut-liver axis plays an important role in the pathogenesis of liver diseases, including liver fibrosis and cirrhosis. Chronic liver disease, especially fibrosis and cirrhosis, is a terrifying disease with many side effects. The fact that liver cirrhosis is related to the microbiome and the possibility that it can be treated by controlling the microbiome is expected to affect the development and health improvement in the medical field in the future. Therefore, it is necessary to evaluate the manipulation of the intestinal microbiota in the context of liver cirrhosis.

Consequently, a comprehensive understanding of the biology of liver cirrhosis is important for improving clinical outcomes, as integrated signaling pathways appear to play an important role in pathogenesis of liver cirrhosis. Further studies are needed to study the interaction between gut microbes and the host immune system in order to elucidate the pathogenesis of liver fibrosis and open new opportunities in immunity or gut microbiome-based treatments. Future trials of probiotics, prebiotics, and synbiotics are recommended to include metagenomic and/or metabolomic analysis for evaluation of their effects and their possible backgrounds.”

  • Comment 3: A fair English revision is needed. 

  • Response 3: As the reviewer pointed out, we commissioned a professional who is good at proofreading English and saw the correction.

Reviewer 2 Report

This work deals with the role of gut microbiome in hepatic diseases until liver cirrhosis, and on their possible treatments with prebiotics and probiotics. The study shows some gaps regarding the link between microbiota, inflammation and chronic liver disease. Inflammation is little considered. Since a review does not have to be based only on a reading of the literature, authors should give their interpretation to the data currently available and take a position on the matter.

Detailed comments:

Page 1: 1) In the abstract, line 9, we suggest replacing the term “genetic conditions” with “diseases”.

Page 2: 2) Line 54: Briefly, clarify was is intended for chronic injury, comment on the possible role of matrix metalloproteinases (MMPs), and describe the state of inflammation if any, as well as the relevance of immunological markers 3) Line 55: As new disease terms are introduced, add corresponding abbreviations. 4) Line 59: “excessive drinking” of what? 5) Lines 67-68: add references. Please note that the ratio microbes/host cells has been reduced to just over one (Sender R, Fuchs S and Milo R (2016) Are we really vastly outnumbered? Revisiting the ratio of bacterial to host cells in human. Cell, 164, 337-340; also read in Riccio P, Rossano R. The human gut microbiota is neither an organ nor a commensal. FEBS Lett. 2020; 594(20):3262-3271. Review. 6) Line 68: “The gut microbiota is responsible for removing invading pathogens…” . Actually, they also prevent the invasion by pathogens.

Page 3: 7) Paragraph on dysbiosis:This paragraph is not correct and incomplete. As a matter of fact, persistent dysbiosis may be associated with persistent intestinal inflammation, then with the opening of the intestinal barrier, endotoxemia and systemic chronic inflammation. Diseases may be gastrointestinal, but also systemic and in particular chronic neuroinflammatory diseases in case of damage of the blood brain barrier (BBB). 8) Line 99: As the “imbalance of the microbiota” in dysbiosis is not precisely defined, can not serve as an indicator for pathological state. 9) Lines 117-118: As new abbreviations are introduced, add corresponding terms. 10) Line 119: Instead of “form” use “represent”… 11) Lines 129-130: please clarify what is meant by "the abundance of major gut microbiota."

Page 4: 12) Line 141: The sentence “Moreover, antibiotics affect the bacteria that cause infections as well as the resident microbiota” and the following one should be better expressed. 13) Line 153: patients with what diseases? 14) Line 156: add some other less dated reference to reference [43]. 15) Lone 167: As new abbreviations are introduced, add corresponding terms.

Page 5: 16) Tables would be better visible if the content were aligned left.

Page 6: 17) Line 208: write prebiotics instead of probiotics.

Page 8: 18) Line 283: add a reference after antibiotics. 19) Line 284: clarify what is meant with “these findings”. 20) Line 291: clarify what is meant with a “healthy diet”. 21) Line 292: please replace a “healthy fecal microbiome” with “fecl microbiome from healthy subjects”. 22) Line 295: add references. 23) Line 301: is the term “pathologic biology” really appropriate?

Author Response

ijms-1041366

“The modulation of the gut microbiome in liver fibrosis”  

Point-to-point responses to comments by the Reviewer 2.

Reviewer 2: Suggestions for Authors This work deals with the role of gut microbiome in hepatic diseases until liver cirrhosis, and on their possible treatments with prebiotics and probiotics. The study shows some gaps regarding the link between microbiota, inflammation, and chronic liver disease. Inflammation is little considered. Since a review does not have to be based only on a reading of the literature, authors should give their interpretation to the data currently available and take a position on the matter.

  • Response : We agree with the reviewer’s comment and mentioned their interpretation to the data currently available and take a position on the matter. When we designed our study, the purpose of this review paper is to provide insight and research direction by collecting data on possible treatments as probiotics, prebiotics, synbiotics, and antibiotics as the role of the gut microbiome in liver cirrhosis. We have further refined our conclusions to interpret the data and take a position on the issue. Thanks again for the many tips and detailed comments. The reviewer's suggestion was of great help in the process of being more accurate and improving our paper.

Major Criticism

  • Detailed comments 1: Page 1: 1) In the abstract, line 9, we suggest replacing the term “genetic conditions” with “diseases”.
  • Response 1: Thanks for correction suggestion. We replaced the sentence as suggested, the interpretation became more accurate.

  • Detailed comments 2: Page 2: 2) Line 54: Briefly, clarify was is intended for chronic injury, comment on the possible role of matrix metalloproteinases (MMPs), and describe the state of inflammation if any, as well as the relevance of immunological markers.
  • Response 2: Thanks for raising this point. We mentioned about possible role of matrix metalloproteinases (MMPs), and describe the state of inflammation if any, as well as the relevance of immunological markers.

“Although liver fibrosis is a local reaction of the liver to chronic injury, serum levels of fibrogenic cytokines, extracellular matrix proteins, and degradation products are markedly increased in cases of advanced fibrosis (bridging fibrosis or cirrhosis) [14]. The matrix metalloproteinases (MMPs) and their inhibitors, tissue inhibitors of metalloproteinases, are proteins important in the matrix degradation. The finding that MMPs are expressed in hepatic injury indicates degradation of normal extracellular matrix may contribute to liver fibrosis. In a previous report, MMPs and microbiome are associated with fibrosis in respiratory disease [15]. MMPs play a central role in extracellular matrix remodeling in normal physiology. Each MMPs are associated with different stages of liver injury, including disease resolution, liver inflammation, fibrosis, cirrhosis and hepatocellular carcinoma.”

  • Detailed comments 3: 3) Line 55: As new disease terms are introduced, add corresponding abbreviations.
  • Response 3: As reviewer pointed out, abbreviations of diseases are attached in parentheses. We added it to following sentence: "The most common causes of cirrhosis are viral hepatitis (HV), nonalcoholic steatohepatitis (NASH), and alcoholic liver disease (ALD) [14]. Hepatocellular carcinoma (HCC) occurs in the background of a cirrhotic liver [15].”

  • Detailed comments 4: 4) Line 59: “excessive drinking” of what?
  • Response 4: Excessive drinking means over drinking of alcohol. For better understanding, we will add the “of alcohol” after “excessive drinking”.

  • Detailed comments 5: 5) Lines 67-68: add references. Please note that the ratio microbes/host cells has been reduced to just over one (Sender R, Fuchs S and Milo R (2016) Are we really vastly outnumbered? Revisiting the ratio of bacterial to host cells in human. Cell, 164, 337-340; also read in Riccio P, Rossano R. The human gut microbiota is neither an organ nor a commensal. FEBS Lett. 2020; 594(20):3262-3271. Review.
  • Response 5: We agree with the reviewer’s comment and apologize for causing confusion. We changed sentence and reviewed all suggested references.

“Most people have a population of bacteria in the gut that is about similar number of cells in the body [21].”

  • Detailed comments 6: 6) Line 68: “The gut microbiota is responsible for removing invading pathogens…” . Actually, they also prevent the invasion by pathogens.
  • Response 6: Thank you for finding the missing part. As you might say, the gut microbiota acts as a deterrent as well as pathogen removal. we added another reference on this, "The gut microbiota is responsible for removing invading pathogens, in addition to maintaining the balance of the immune system and preventing autoimmunity. We modified it to the following sentence: "Corrected the sentence to “The gut microbiota is responsible for preventing and eliminating the invasion of pathogens, in addition to maintaining the balance of the immune system and preventing autoimmunity.”

  • Detailed comments 7: Page 3: 7) Paragraph on dysbiosis: This paragraph is not correct and incomplete. As a matter of fact, persistent dysbiosis may be associated with persistent intestinal inflammation, then with the opening of the intestinal barrier, endotoxemia and systemic chronic inflammation. Diseases may be gastrointestinal, but also systemic and in particular chronic neuroinflammatory diseases in case of damage of the blood brain barrier (BBB).
  • Response 7: Thanks for pointing out. We also thought that this part was insufficient to explain, so we supplemented it with the results of previous studies.

“For this reason, many studies are being performed to investigate the possibility of alleviating cirrhosis by modulating the gut microbiome. Dysbiosis of the gut is related with various human diseases such as obesity [40], metabolic disease [41], diabetes mellitus [42], vascular disease [43], chronic liver disease [44, 45] and brain disease [46]. Neuropsychologically, bidirectional interaction (gut-brain axis) between brain and gut microbiome via neurological or immunological mechanisms is closely related with dysbiosis.”

  • Detailed comments 8: 8) Line 99: As the “imbalance of the microbiota” in dysbiosis is not precisely defined, can not serve as an indicator for pathological state.
  • Response 8: Thank you for pointing out the insufficient explanation of it. In order to explain it concretely, it has been modified as follows: Dysbiosis is a term for a microbial imbalance or maladjustment inside the body. This state of imbalance can be caused by changes in the number of microbes in a particular population that can have a profound influence on energy and immune homeostasis, which result in significant metabolic and immunologic effects on the host [33]. This persistent imbalance of gut microbial community is accompanied by a wide range of systemic symptoms of gastrointestinal diseases such as inflammatory bowel disease and irritable bowel syndrome, obesity, type 2 diabetes, and atopy [34].

  • Detailed comments 9: 9) Lines 117-118: As new abbreviations are introduced, add corresponding terms.
  • Response 9: There is a lack of points that have not been thoroughly checked. Thanks for pointing out this. We added corresponding terms of new abbreviations: alanine aminotransferase (ALT) and triglycerides (TG).

  • Detailed comments 10: 10) Line 119: Instead of “form” use “represent”…
  • Response 10: Thanks for raising this point. As you told, we modified the confused meaning to the exact meaning. “represent” gives more clear meaning.

  • Detailed comments 11: 11) Lines 129-130: please clarify what is meant by "the abundance of major gut microbiota."
  • Response 11: Previous study showed the association between gut microbiota in cirrhosis. As liver cirrhosis progressed, the ratio of Enterobacteriaceae, Lachonospiraceae, Ruminococcaceae and Blautia was lower than that of the control group. We expressed to say that this changes the abundance of major gut microbiota. However, since there was no specific explanation, we modify it as the following sentence: “It suggests that cirrhosis is associated with a reduced conversion of primary to secondary Bas, which is associated with the abundance of major gut microbiota such as Enterobacteriaceae, Lachonospiraceae, Ruminococcaceae and Blautia.”

  • Detailed comments 12: Page 4: 12) Line 141: The sentence “Moreover, antibiotics affect the bacteria that cause infections as well as the resident microbiota” and the following one should be better expressed.
  • Response 12: According to your advice, it is expressed additionally as the following sentence: Antibiotics are used with the intention of removing pathogenic bacteria and inhibiting proliferation, but due to their broad-spectrum activity, they can indiscriminately kill or inhibit a subset of symbiotic microbes.

  • Detailed comments 13: 13) Line 153: patients with what diseases?
  • Response 13: According to the two references mentioned, we revised to describe "patients with non-alcoholic fatty liver disease and irritable bowel syndrome" in detail.

  • Detailed comments 14: 14) Line 156: add some other less dated reference to reference [43].
  • Response 14: As the you said, we have added a recent reference related to the sentence.: Shin, D.; Chang, S. Y.; Bogere, P.; Won, K.; Choi, J. Y.; Choi, Y. J.; Lee, H. K.; Hur, J.; Park, B. Y.; Kim, Y.; Heo, J., Beneficial roles of probiotics on the modulation of gut microbiota and immune response in pigs. PLoS One 2019, 14, (8), e0220843. This paper also shows the results of proliferation of beneficial bacteria and inhibition of harmful bacteria by controlling the intestinal microflora due to the intake of Lactobacillus-based probiotics.

  • Detailed comments 15: 15) Lone 167: As new abbreviations are introduced, add corresponding terms.
  • Response 15: Thank you for pointing the part we missed. New abbreviations have been resolved and written as “collagen type 1a (Col1a), tissue inhibitor of metallopeptidase 1 (Timp1)”

  • Detailed comments 16: Page 5: 16) Tables would be better visible if the content were aligned left.
  • Response 16: As you suggested, the content of the Main Results is aligned to the left. And adjusted the table a little to look more visible.

  • Detailed comments 17: Page 6: 17) Line 208: write prebiotics instead of probiotics.
  • Response 17: Thank you for pointing out a small mistake that we haven't yet identified. Corrected "probiotics" typo to "prebiotics".

  • Detailed comments 18: Page 8: 18) Line 283: add a reference after antibiotics.
  • Response 18: As you suggested, we added a reference that modifying the diet by regulating the intestinal environment caused by antibiotics, probiotics, and prebiotics can also provide significant benefits for patients with cirrhosis hepatic encephalopathy. “Campion, D.; Giovo, I.; Ponzo, P.; Saracco, G. M.; Balzola, F.; Alessandria, C., Dietary approach and gut microbiota modulation for chronic hepatic encephalopathy in cirrhosis. World J Hepatol 2019, 11, (6), 489-512.”

  • Detailed comments 19: 19) Line 284: clarify what is meant with “these findings”.
  • Response 19: As the reviewer said, what these findings point to is ambiguous, so we supplemented the sentence with: “The pathogenesis of cirrhosis and the precise function of gut microbiome are not yet clear, but these findings that improvement in liver cirrhosis-induced animals and patients highlight the importance of modulation of the gut microbiome, suggesting novel approaches for therapeutic strategies for liver fibrosis.”

  • Detailed comments 20: 20) Line 291: clarify what is meant with a “healthy diet”.
  • Response 20: A healthy diet refers to prebiotics or dietary fiber-based, as it helps the activity of beneficial bacteria in the gut. In this regard, the sentence was revised as follows: "A recent review revealed that the modulation of gut microbiota with a healthy diet that helps gut microbial activity such as fiber-based, metabiotic-based supplements, and transplantation of a fecal microbiome from healthy subjects to promote the growth of “good” microbiota may ameliorate dysbiosis in patients and improve their prognosis.”

  • Detailed comments 21: 21) Line 292: please replace a “healthy fecal microbiome” with “fecl microbiome from healthy subjects”.
  • Response 21: Thanks for the reviewer's advice. This helped in natural interpretation. The sentence was amended as follows: "A recent review revealed that the modulation of gut microbiota with a healthy diet that helps gut microbial activity such as fiber-based, multi-biotics based supplements, and transplantation of a fecal microbiome from healthy subjects to promote the growth of “good” microbiota may ameliorate dysbiosis in patients and improve their prognosis.”

  • Detailed comments 22: 22) Line 295: add references.
  • Response 22: We are sorry to confuse you by citing without reference. As your suggestion, we marked the reference follow as: “Ribeiro, C. F. A.; Silveira, G. G. d. O. S.; Cândido, E. d. S.; Cardoso, M. H.; Espínola Carvalho, C. M.; Franco, O. L., Effects of Antibiotic Treatment on Gut Microbiota and How to Overcome Its Negative Impacts on Human Health. ACS Infectious Diseases 2020, 6, (10), 2544-2559.”

  • Detailed comments 23: 23) Line 301: is the term “pathologic biology” really appropriate?
  • Response 23: We have made this expression because several studies have confirmed pathological improvement in animal laboratory results, but for clarity we will amend the term "biology".

Reviewer 3 Report

The authors Lee et al. have reviewed a wide breath of literature on liver cirrhosis and potential microbiome-related therapeutics. The paper is well-constructed and written. That said, I would like the authors to described about dysbiosis section to standout from already available papers. The section need more information and possibly a figure would be beneficial.

Gut-liver axis is a hot-spot for maintaining immune system. Perhaps, the authors should talk more about the intestinal barrier dysfunction, microbial translocation (MT) and potential immune implication. For example, MT has been correlated to Gram-negative bacteria such as Proteobacteria phylum.

All the 16s gene seq information limits us to genus-level. Future studies should focus on metagenomics where we can reach up to species-level of microbes and, perhaps modulating with pre-pro or syn-biotics after looking at species-level will provide more benefit. It would be beneficial to add a brief section on this.

Author Response

ijms-1041366

“The modulation of the gut microbiome in liver fibrosis”  

Point-to-point responses to comments by the Reviewer 3.

Reviewr 3: The authors Lee et al. have reviewed a wide breath of literature on liver cirrhosis and potential microbiome-related therapeutics. The paper is well-constructed and written. That said, I would like the authors to described about dysbiosis section to standout from already available papers. The section need more information and possibly a figure would be beneficial.

  • Response 14: Thank you for your deep insight and advice. As your advice, we supplemented the dysbiosis section. We would like to make the figure as suggested by the reviewer to help clear understanding, but the results are different for each study, so it was difficult to create a single unified figure. However, to help you understand more easily, a figure that summarizes the integrated contents is attached as follows.

Figure 1. Dysbiosis and diseases

AST, aspartate aminotransferase; ALT, alanine aminotransferase; T-BIL, total bilirubin; ALB, albumin; MDA, malondialdehyde; SOD, superoxide dismutase; GSH, Glutathione; CTP, Child-Turcotte-Pugh; MELD, model for end-stage liver disease; Col, Collagen; Timp, tissue inhibitor of metallopeptidase; TGF, transforming growth factor; α-SMA, alpha-smooth muscle actin; TNF- α, tumor necrosis factor alpha; Zo-1, zonula occludenes-1; BA, bile acids; BCL-2, b-cell lymphoma 2

Major Criticism

  • Comment 1: Gut-liver axis is a hot-spot for maintaining immune system. Perhaps, the authors should talk more about the intestinal barrier dysfunction, microbial translocation (MT) and potential immune implication. For example, MT has been correlated to Gram-negative bacteria such as Proteobacteria phylum.
  • Response 1: Thanks for these suggestions. We mentioned about MT on dysbiosis section.

“In fibrosis and cirrhosis, intestinal dysbiosis, gut barrier dysfunction, and systemic immunologic dysfunction cause bacterial translocation [53]. Bacterial translocation is defined as the migration of viable intestinal microorganisms or their products to the mesenteric lymph nodes or other sites. The liver is a central immunological organ that is composed with innate immune cells and constantly exposed to circulating endotoxins derived from intestinal microbiome [54]. Intestinal immune dysregulation due to intestinal immune system abnormalities is main event in patients with cirrhosis. With cirrhosis progression, intestinal immune dysregulation and gut dysbiosis worsened.”

  • Comment 2: All the 16s gene seq information limits us to genus-level. Future studies should focus on metagenomics where we can reach up to species-level of microbes and, perhaps modulating with pre-pro or syn-biotics after looking at species-level will provide more benefit. It would be beneficial to add a brief section on this.
  • Response 2: As the reviewer pointed out, we added sentences on ‘Cirrhosis and Gut Microbiome’ section.

“Most of the studies used 16S rRNA sequencing, and detailed analysis up to the genus level was possible. Future studies should focus on metagenomics where we can reach up to species-level of microbes by using total DNA sequencing (shotgun metagenomics). And then, we can identify and characterize biological mechanisms that drive the human response to an intervention of pharmabiotics.”

Round 2

Reviewer 1 Report

The manuscript has been improved. However, a English native revision is needed. 

Reviewer 2 Report

The authors accepted and responded to all the points made in the first review. However, all changes need to be checked for English, especially the sentences starting at lines 63, 186, 387, 402. We particularly appreciate the new title.
Some comments in detail:
1) in line 150, we would say "neuroinflammatory diseases" instead of "brain disease";
2) and at line 420, we would say "pathology" instead of "biology".

Regarding the number of microbes being slightly greater than the number of human cells, we would recommend reading the work of Sender et al. (Cell, 2016) and citing it.

Author Response

Response to Reviewer 2 Comments

Comment: The authors accepted and responded to all the points made in the first review. However, all changes need to be checked for English, especially the sentences starting at lines 63, 186, 387, 402. We particularly appreciate the new title.

Response: With your point and advice, our paper has been greatly improved. Thank you again for your insight and attention. The sentences pointed out were once again reviewed in English. Sorry for not being smooth English. The revised the title makes our thesis better expressed. Thank you.

Point 1: 1) in line 150, we would say "neuroinflammatory diseases" instead of "brain disease"

Response 1: Thanks for the good point. Brain disease was a too comprehensive expression. It was corrected as " neuroinflammatory disease".

Point 2: 2) and at line 420, we would say "pathology" instead of "biology".

Response 2: We modified “biology” to “pathology” as your advice. It became a more flexible sentence.

Point 3: Regarding the number of microbes being slightly greater than the number of human cells, we would recommend reading the work of Sender et al. (Cell, 2016) and citing it.

Response 3: Thank you for detailed description. Sender et al. (Cell, 2016) as referred to the paper, we corrected the relevant sentence, and cited it again.

[After modification]

Most people have a population of bacteria in the gut that is about 10-factor number of cells in the body [21].
